# Notch family members follow stringent requirements for intracellular domain dimerization at sequence-paired sites

**Jacob J. Crow**[1], **Allan R. Albig**[1,2]*

**1** Biomolecular Sciences PhD Program, Boise State University, Boise, ID, United States of America,
**2** Department of Biological Sciences, Boise State University, Boise, ID, United States of America

* AllanAlbig@boisestate.edu

**Data Availability Statement:** All relevant data are within the manuscript and its Supporting Information files.

**Funding:** This work was supported by funding from the National Institute of General Medical

## Abstract

Notch signaling is essential for multicellular life, regulating core functions such as cellular identity, differentiation, and fate. These processes require highly sensitive systems to avoid going awry, and one such regulatory mechanism is through Notch intracellular domain dimerization. Select Notch target genes contain sequence-paired sites (SPS); motifs in which two Notch transcriptional activation complexes can bind and interact through Notch's ankyrin domain, resulting in enhanced transcriptional activation. This mechanism has been mostly studied through Notch1, and to date, the abilities of the other Notch family members have been left unexplored. Through the utilization of minimalized, SPS-driven luciferase assays, we were able to test the functional capacity of Notch dimers. Here we show that the Notch 2 and 3 NICDs also exhibit dimerization-induced signaling, following the same stringent requirements as seen with Notch1. Furthermore, our data suggested that Notch4 may also exhibit dimerization-induced signaling, although the amino acids required for Notch4 NICD dimerization appear to be different than those required for Notch 1, 2, and 3 NICD dimerization. Interestingly, we identified a mechanical difference between canonical and cryptic SPSs, leading to differences in their dimerization-induced regulation. Finally, we profiled the Notch family members' SPS gap distance preferences and found that they all prefer a 16-nucleotide gap, with little room for variation. In summary, this work highlights the potent and highly specific nature of Notch dimerization and refines the scope of this regulatory function.

## Introduction

Notch signaling is a cornerstone of multicellularity and dictates cellular fate and identity. Notch signaling is heavily influenced by microenvironmental cues [1], including adjacent "sending cells" which present any of five Notch ligands to up to four Notch receptors expressed on so called "receiving cells". Ligand bound and activated Notch receptors undergo a series of proteolytic cleavages which release an active intracellular domain (NICD) [2–4]. This transcriptionally active fragment translocates to the nucleus to act as a co-transcription factor.

Sciences (https://www.nigms.nih.gov) to A. Albig (2R15GM102852-02 and 1R15GM134501-01) and from NIGMS grants P20GM103408 and P20GM109095.

**Competing interests:** The authors have declared that no competing interests exist.

Common Notch signaling targets are transcription factors themselves, which have their own broader implications and cascades, culminating in a system that requires a fine-tuned, highly sensitive signaling network. Disruption of Notch signaling, both through over- and under-activation, leads to a variety of developmental abnormalities and cancers [5]. Understanding mechanisms behind this precise level of internal control may pave the way for treatments of many of its resulting disorders.

The mammalian Notch signaling system consists of four mostly homologous receptors (Notch1-4) which are all activated through this manner. Each NICD molecule can be readily split into three sections; the N-terminus which contains the RBPJ associated module (RAM) domain, the central ankyrin domain, and a variable C-terminus which houses the Pro-Glu-Ser-Thr (PEST) domain used in protein turnover and in some Notch proteins, a transactivation domain (TAD). Through the RAM domain, all Notch proteins bind to the same transcription factor, recombination signal binding protein for immunoglobulin kappa J region (RBPJ, also commonly called CSL, CBF-1/Suppressor of Hairless/Lag-1) [6]. Upon NICD binding to RBPJ, a new NICD/RBPJ interface is formed which recruits another co-activator, a member of the Mastermind-like (MAML) family [7]. This new tri-protein complex recruits a cascade of other transcriptional machinery to drive transcription of its target genes [8]. While each Notch protein contains the conserved RAM and ankyrin domains, their transcriptional activation profiles are not identical and are largely dependent on context within promoter elements [9].

The DNA target sites which the Notch transcriptional activation complex (NTC) binds to have been the subject of thorough analysis. The consensus binding site was originally defined as a "TP1 element" with the sequence 5′ CGTGGGAAAAT 3′ that recruits RBPJ to Notch responsive promoters [10,11]. TP1 elements are found in a variety of configurations within promoters. Perhaps most importantly, TP1 elements orientated in a head-to-head directionality and separated by 16 base pairs (bp), also known as sequence-paired sites (SPS), enable cooperative binding of two NICD molecules [12,13]. This cooperation results in better repression in the absence of NICD, and enhanced activation in its presence [9,14]. Upon modeling of two N1ICD transcriptional cores on a SPS, it was proposed that complex dimerization occurs through the N1ICD ankyrin domain [15] and this was further supported through crystallization of the interface [16]. Importantly, theses SPS-driven promoters appear to be dimer-dependent. When dimerization was interrupted, N1ICD's transcriptional potential was substantially reduced on promoters containing SPSs [15,16] and could no longer induce T-cell acute lymphoblastic leukemia [17]. Together, sequence-paired sites and Notch dimerization appear to be potent regulators of Notch signaling and warrant a closer investigation into their mechanics.

In the search for new Notch responsive genes, ChIP-Seq approaches have recently been adopted to identify new SPS sites based on DNA interaction with RBPJ. While the NTC-dimer crystal structure dictates a 16-nucleotide spacer region, ChIP-Seq analysis by Castel, et al. identified a variety of potential SPS-driven genes with spacer regions from 11 to 21 base pairs [18]. These possible targets are opposed by *in vitro* analysis which observed a more limited spacer region of 15 to 17 base pairs [15,19]. This discrepancy in spacer length is further complicated due to ChIP-Seq approaches that experimentally identified individual RBPJ binding sites then computationally screened for nearby secondary sites [18–20]. Screening for secondary sites however is not straightforward since loading of a NTC onto a high-affinity site directly enables cooperativity on cryptic, low affinity sites which may not even resemble traditional RBPJ binding sequences [16,17,19]. While the transcriptional outcomes seem to be clear, the mechanisms dictating this SPS-response within promoters and enhancers are not clearly understood.

While dimerization-induced signaling of Notch1 has been previously explored, the ability, specifications, and limitations for the other members of the Notch family remain unknown.

To compare dimer-dependent signaling of the various NICDs, we generated luciferase reporter constructs driven by either isolated sequence-paired sites from known dimer-dependent promoters or an artificial/optimized SPS site. We observed that all NICDs activate these promoters with varying efficiency. We also observed that Notch1, 2, and 3 functions through dimerization dependent mechanisms, while Notch4 appeared dimer independent. Finally, we compared the optimal gap length within SPS sites and found that all NICD molecules prefer promoters with 16bp between RBPJ binding sites, with little room for deviation. These results should help us to understand how the various NICD molecules interact in cells and potentially diversify Notch signaling outputs in cells that express multiple Notch proteins.

## Materials and methods

### Cell culture

HEK293T cells were cultured in Dulbecco's Modified Eagle's Medium (DMEM) (Mediatech, Inc.) supplemented with 10% EquaFetal Bovine Serum (FBS) (Atlas Biologicals) and 1x penicillin-streptomycin solution (Mediatech, Inc.). Cells were grown in 10 cm culture plates and subcultured at 70–80% confluency.

### Expression and reporter plasmids

Protein expression constructs were obtained through the following: FLAG-N1ICD (AddGene #20183), N2ICD (#20184), N3ICD (#20185), and N4ICD (#20186) were all gifted by Raphael Kopan [9] and acquired through AddGene.org. All constructs code for the intracellular domain of the mouse Notch proteins and have a 3xFLAG peptide tag on the N-terminus. The N1ICD-MYC ΔS2184 construct, also a gift from Raphael Kopan [2] (#41730), includes a substantial C-terminal truncation, encoding mouse N1ICD V1744 to S2184 with a MYC tag located at the C-terminus. The NICD coding regions were subcloned into pKH3 (#12555), a gift from Ian Macara [21] to add a C-terminal 3xHA tag. N1ICD (R1974A), N2ICD (R1934A), N3ICD (R1896A), and N4ICD (R1685A) mutants were all created through site-directed mutagenesis of the NICDs based on sequence alignment to identify amino acids (S2 Fig) in mouse NICDs homologous to the human N1CID R1984 site previously shown to be essential for NICD dimerization [16]. The empty coding vector pcDNA3.1/MYC-His was obtained from Invitrogen and pCMV-β-Galactosidase was obtained from Clontech/Takara Bio.

Transcriptional reporter constructs were obtained or created as the following: Full-length mouse promoters for *Hes1* (#41723) and *Hes5* (#41724) were a gift from Ryoichiro Kageyama and Raphael Kopan [22]. 4xTP1 (#41726), a synthetic promoter containing four high-affinity RBPJ binding sites in tandem, was a gift from Raphael Kopan [23]. These promoter sequences were designed and cloned into pGL2-Basic (Promega), a luciferase reporter plasmid, which upon promoter activation drives expression of firefly luciferase.

### Construct creation and mutagenesis

To create luciferase reporters that activate specifically upon Notch dimerization, we isolated the sequence-paired sites from the native mouse and human *Hes1* and *Hes5* genes and cloned these fragments into the pGL3-Basic vector (Promega) which contains a minimal promoter that is incapable of transcriptional initiation without additional enhancer elements. Our synthetic promoters, the 2xTP1 constructs, were designed using the TP1 response element as originally isolated from the Epstein-Barr virus, which contains two high-affinity binding site for RBPJ [10,11]. TP1's 'complete' RBPJ consensus sequence (5'-CGTGGGAAAAT-3') and a ubiquitous "core" sequence (5'-GTGGGAA-3') were taken from the response element, its

secondary site reversed and placed in the complementary strand to result in a head-to-head arrangement, and two nucleotides were inserted into TP1's spacer region to result in a sixteen-base pair, sequence-paired site.

These fragments were synthesized as oligonucleotides (Integrated DNA Technologies, IDT) and were designed to be partially complementary so that when annealed, the ends were left overhanging, matching cuts from the restriction enzymes KpnI and SacI (New England Biolabs, NEB). pGL3-Basic was cut with these two enzymes, dephosphorylated with shrimp alkaline phosphatase (NEB), inserts phosphorylated with T4 Polynucleotide Kinase (NEB), and ligated together with T4 DNA Ligase (NEB).

A variety of sequence-paired site gap distances were created through blunt-end ligation. PCR primers to mutate Hes1, Hes5, and 2xTP1 were designed to align with the desired base excisions or to include base extensions. Base extensions were designed to keep the gap distance nucleotide composition (G/C vs A/T) approximately consistent with the native promoters'. PCR products were phosphorylated, ligated, and reaction template digested with the restriction enzyme DpnI (NEB). A statistical comparison of the modified constructs' basal activity levels did not indicate the addition or subtraction of any other regulatory elements (data not shown).

All constructs were sequenced verified before experimental use. SPS sequence information can be found in S1 Table.

## Western blotting

For western blotting analyses, HEK293T cells were plated into 6-well plates at a density of 300,000 cells/well. The following day, cells were transfected with polyethylenimine MW 25,000 (PEI, Polysciences) at a ratio of 5 μg PEI to 1 μg DNA. Wells were transfected with 1000 ng of plasmid DNA for the various FLAG-NICD constructs, allowed to grow for two days, and cells collected and prepared in 1x SDS-page lysis buffer. Western blotting was performed as described previously [24]. Membranes were incubated with primary antibodies against FLAG tag (Cell Signaling Technology, #14793) or GAPDH (Santa Cruz Biotechnology, sc-25778) and detected through horseradish peroxidase conjugated α-rabbit antibodies (GE Healthcare Life Sciences, NA934V). Experiments were repeated independently three times, where the figure displayed uses the best representative exposures.

## Luciferase assays

For all luciferase assays, HEK293T cells were plated into 24 well plates at a density of 50,000 cells/well where the experimental conditions were treated as triplicates or duplicates. The following day, cells were transfected. When analyzing the full-length promoters, 100 ng of luciferase construct and 10 ng of NICD expression plasmid were used, whereas in experiments of sequence-paired site constructs, 200 ng and 100 ng were used, respectively. In all experimental variations, 10 ng of a β-Galactosidase expression plasmid was used per well to normalize data for transfection efficiency and cell growth/death. To equate amounts of DNA between experimental conditions, the empty coding plasmid pcDNA3.1/MYC-His was utilized. Forty-eight hours post transfection, cells were collected and analyzed as previously described [24]. All samples were treated in triplicates, except for the single-base change experiment, which was in duplicates. Independent experiments were performed at least four times.

## Statistical analysis

Statistical significance was determined through a student's two-tailed $t$ test, comparing two-samples with homoscedastic variance. Significance is determined as *** is $p \leq 0.001$, ** is $p \leq 0.01$, and * is $p \leq 0.05$.

## Results

### Activation of Notch target genes containing sequence-paired sites requires ankyrin-dependent dimerization

Notch target genes often have multiple RBPJ binding sites within their promoter sequences and a fraction of these are orientated in head-to-head, paired sites [18,19]. This arrangement allows for NTC dimerization through NICD ankyrin domains, resulting in potent transcription of SPS containing genes. The canonical Notch target genes *Hes1* and *Hes5* have previously shown to be activated by Notch1 in a dimer-dependent manner [16,19]. Using luciferase reporter assays, we first sought to confirm if other members of the mammalian Notch family also activate *Hes1* and *Hes5* in a dimer-dependent manner.

HEK293T cells were co-transfected with commercially available luciferase reporter plasmids containing large fragments of the Hes1 or Hes5 promoters, and either wild-type or dimerization incompetent Notch1–4 NICD expression plasmids to observe dimerization dependence of these promoters. In agreement with previous work, the full-length Hes1 (Fig 1A) and Hes5 (Fig 1B) constructs were significantly activated by wild-type N1ICD and

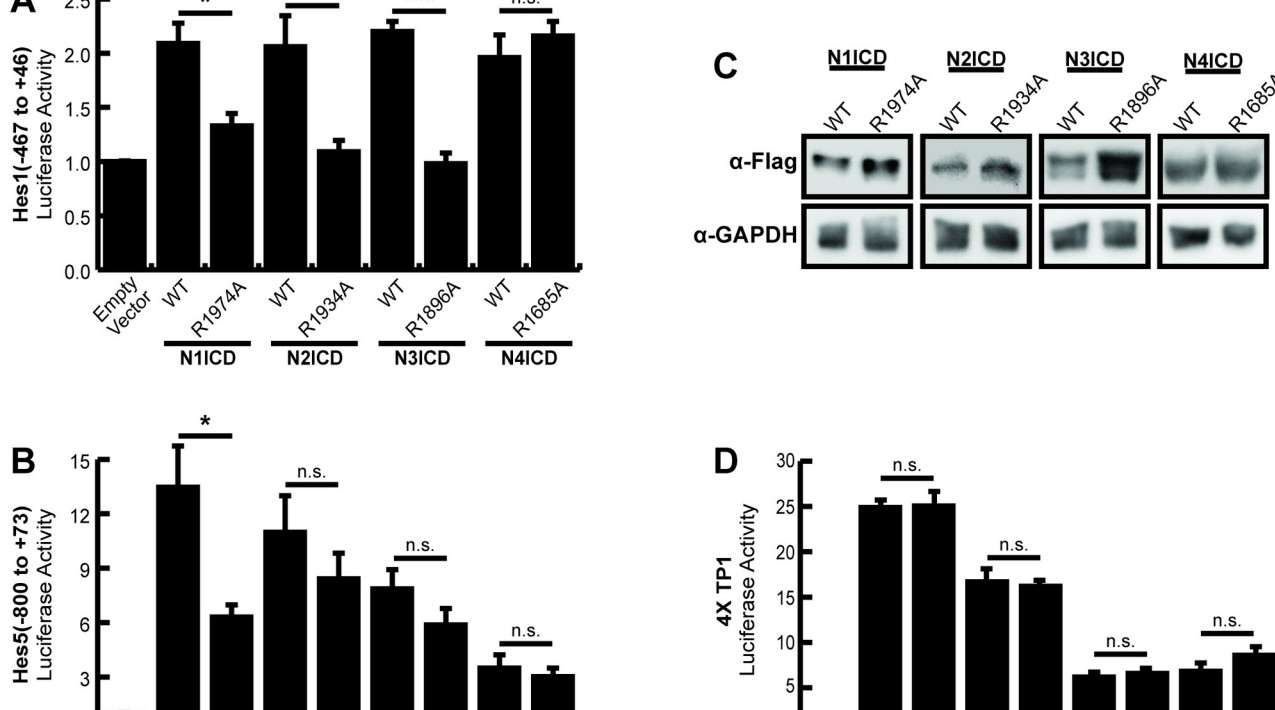

**Fig 1. Dimerization dependence of full-length Hes1 and Hes5 promoters.** HEK293T cells were transfected with portions of the (**A**) *Hes1* (-467 to +46) (**B**) *Hes5* (-800 to +73) promoters for luciferase assays to monitor promoter activation. Cells were co-transfected with wild-type (WT) or ankyrin mutated (R→A) NICDs to compare their dimer dependence. Shown are the average +/- SE of 5 independent experiments. * indicates P < .05, ** < .01, *** < .001 (student's t-test). (**C**) Western blots to compare the protein abundancies of WT or dimer incompetent ankyrin mutated NICDs. HEK293T cells were transfected with NICD expression constructs and NICD proteins were detected with anti-Flag antibodies. Endogenous GAPDH was also detected from the same lysates for use as a loading control. The image displayed is a representative blot from an experiment that was replicated three independent times. (**D**) 293T cells were transfected with the dimer independent 4XTP1 luciferase construct and WT or ankyrin mutated NICDs as above. Shown are the average +/- SE of three independent experiments.

transcriptional activation was significantly less for dimer incompetent N1ICD. Likewise, wild-type N2ICD and N3ICD also displayed activation on both promoters but dimer-dependent activation only on the Hes1 promoter. Indeed, mutant N2ICD and N3ICD dimer incompetent constructs demonstrated a similar ability to activate the Hes5 promoter suggesting the N2ICD and N3ICDs were functioning in a dimer-independent manner on the Hes5 promoter. Finally, N4ICD activated the Hes1 promoter to a similar degree as the other NICDs but was significantly weaker on the Hes5 promoter compared to the other NICDs. Moreover, N4ICD failed to demonstrate dimer dependence on either the Hes1 or Hes5 promoters. In agreement with previous observations [15], western blot analysis of the various NICDs demonstrated that the decreased transcriptional activation of the dimer incompetent ankyrin mutant NICD molecules was not associated with decreased protein expression of these mutant NICDs (Fig 1C). Moreover, mutation of NICD ankyrin domains did not impinge the basal transcriptional activity of NICD molecules since WT and ankyrin mutant NICDs exhibited nearly identical transcriptional activity on the non-dimerizing 4xTP1 promoter (Fig 1D).

Collectively, these results largely supported the previously reported dimer-dependent nature of the Hes1 and Hes5 promoters. However, we also noted several weaknesses with these experiments. In particular, we noticed significant background with modest activation of the Hes1 promoter and reduced dimer dependence for N2-4ICD on the Hes5 promoter. Based on these weaknesses, we more closely examined these promoter sequences (S1 Fig). We searched for high affinity binding sites that were defined by the RBPJ consensus sequence TGTGGGAA, and low affinity sites that were defined as having up to two nucleotide differences compared to the high-affinity sites. We found a multitude of possible low affinity RBPJ binding sites in addition to the high-affinity SPS sites previously described [9]. Based on this result, we hypothesized that the overall promoter complexity and number of potential low affinity RBPJ binding sites within these promoter sequences might have been responsible for the high background, low relative activation, and dimer independence observed in our experiments. Moreover, beyond Notch signaling, Hes genes are also controlled by a variety of other transcription factor families [25,26], including self-regulation [27,28], suggesting that these relatively large promoter fragments might have been responding to both Notch specific and non-Notch transcriptional activity. Collectively, these observations prompted us to perform further experiments on the Hes1 and Hes5 promoters with the goal of refining these promoters into more specific tools to monitor dimer-dependent NICD activation.

## The isolated Hes5 sequence-paired site does not respond to dimerization

Our results in Fig 1 show that the Hes5 promoter demonstrated both stronger activation than the Hes1 promoter and N1ICD dimer dependence. We therefore rationalized that isolation of the Hes5 SPS site should yield a promoter element that would specifically and robustly respond to NICD dimerization. We isolated the mouse and human Hes5 SPS elements containing two head-to-head orientated RBPJ binding sites separated by a 16-nucleotide gap, and the 4 to 5 surrounding nucleotides (Fig 2A) as previously described [16] and cloned these sequences into a luciferase reporter vector containing a minimal promoter that was incapable of transcriptional initiation without additional enhance elements. Interestingly, this cloned region includes one canonical RBPJ binding site and a partnered "cryptic site" which doesn't match standard RBPJ binding sequences, but instead was hypothesized to form the partner site for the high-affinity site [16]. In addition, there are two nucleotide differences between the mouse and human genes in this region (Fig 2A), one of which is inside the cryptic RBPJ binding site. As shown in Fig 2B, we found that both mouse and human isolated Hes5 SPS promoters were indeed responsive to N1ICD, however, neither of these isolated elements appeared to be

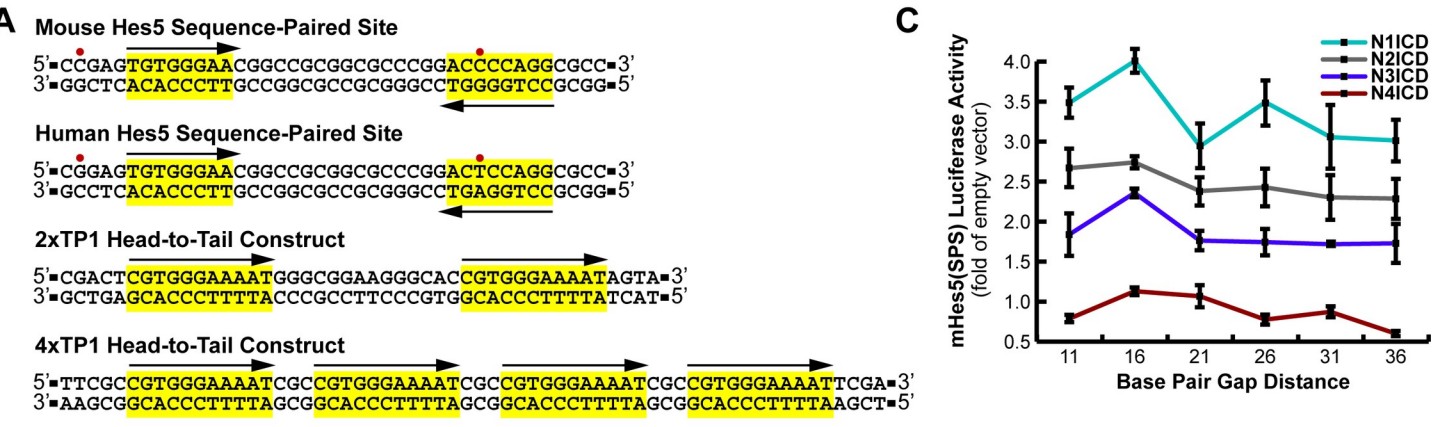

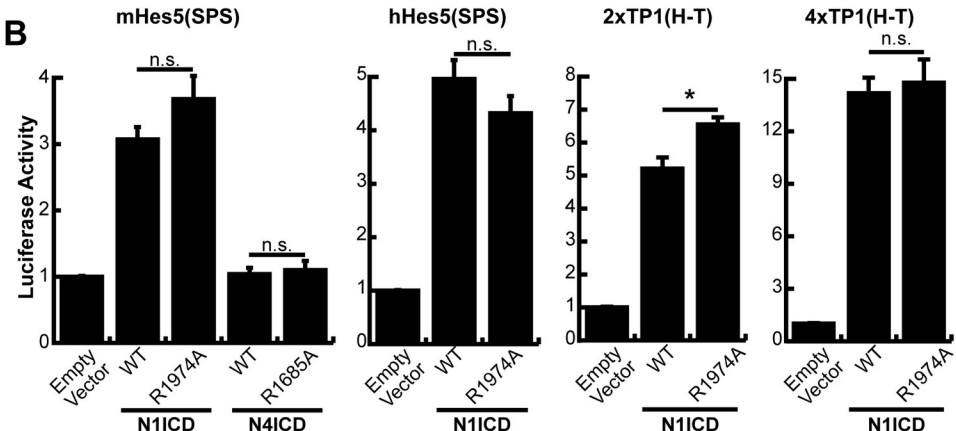

**Fig 2. Lack of dimerization dependence of the *Hes5* sequence-paired site.** (**A**) Promoter structure of the Hes5, 2xTP1(H-T), and 4xTP1(H-T) luciferase constructs. Red dots indicate nucleotide differences between the human and mouse Hes5 SPS sites. Suspected RBPJ binding sites are highlighted in yellow and arrows depict directionality of RBPJ binding sites. (**B**) HEK293T cells were transfected with either the mouse (mHes5(SPS)) or human (hHes5(SPS)) sequence-paired site constructs or with the 2xTP1 (H-T) or 4xTP1(H-T) head-to-tail constructs and WT or dimer-incompetent versions of NICDs. Shown is the average +/- SE of n = 4 independent experiments. * <0.05, ** <0.01, *** <0.001, student's t-test. (**C**) HEK293T cells were transfected with various NICDs and Hes5 SPS constructs with 11–36 bp spacer regions (5 bp increments), n = 6.

dependent on NICD dimerization since they were equivalently activated by WT and dimer-incompetent versions of N1ICD. In addition, the isolated Hes5 SPS responded to Notch activation almost exactly the same as to promoters with RBPJ binding sites in a non-dimerizing head-to-tail orientation (*i.e.* 2xTP1(H-T) and 4xTP1(H-T) constructs). Moreover, N1ICD activated the Hes5 SPS construct less than the 2xTP1 construct despite these promoters both containing two RBPJ binding sites. Interestingly, N4ICD was unable to activate transcription from the isolated Hes5 SPS sites despite successfully activating the full length Hes5 promoter in Fig 1. In stark contrast to this result, the full-length Hes5 promoter did display dimer dependence to N1ICD (Fig 1B). This difference between the full-length and SPS versions of the Hes5 promoter suggested that the Hes5 promoter might have been functioning differently than previously thought. Given the weakness of the cryptic RBPJ binding site, the presence of several other potential RBPJ binding sites within the full-length promoter, and the dimer dependence of the full-length Hes5 promoter, we hypothesized that the high affinity RBPJ binding site might be making a long-distance interaction with another high-affinity site within the promoter. This possibility would allow cooperation between previously thought independent

NTCs within gene promoters and enhancers and perhaps explain the transcriptional differences between full-length and SPS versions of the Hes5 promoter. To test this possibility, we designed a series of Hes5 SPS mutants based on the helical nature of DNA. Since a DNA helix completes approximately one rotation every 10 bp, we hypothesized that an insertion or deletion of five extra base pairs would place the secondary NTC on the opposite side of the DNA, breaking any cooperativity. Further, an addition of an extra five nucleotides may restore cooperativity, despite the NTC being further away, due to a long range NTC dimerization. However, no obvious cooperative binding, or loss thereof, was observed in these constructs (Fig 2C).

Collectively, these results indicated that the Hes5 SPS does not function in a dimer-dependent manner in cells and instead behaves as a monomeric RBPJ binding site. Due to this unexpected complication, to further explore SPS capabilities we moved on to the Hes1 promoter, which contains a more canonical SPS site.

### The Hes1 SPS acts through traditional NICD dimerization mechanisms

*Hes1* is another Notch target gene which demonstrates dimer-dependence. Its promoter contains a single canonical RBPJ binding site and secondary site 16 nucleotides away in the reverse orientation (Fig 1A). This secondary site is slightly non-consensus, but only displays a minor decrease in RBPJ affinity compared to the high affinity consensus site [19,29]. Therefore, we took the same reductionist approach used with *Hes5* and isolated the SPS element out of the Hes1 promoter.

Since the human and mouse sequences in this promoter region are identical, a generic Hes1(SPS) construct was cloned into a minimal promoter luciferase vector to quantitatively monitor its activation by Notch signaling (Fig 3A, Top). As before, transfections of this construct into HEK293T cells, along with WT or ankyrin mutated NICDs, allowed us to analyze dimer dependence of the Hes1 SPS element. The Hes1(SPS) construct responded to NICD

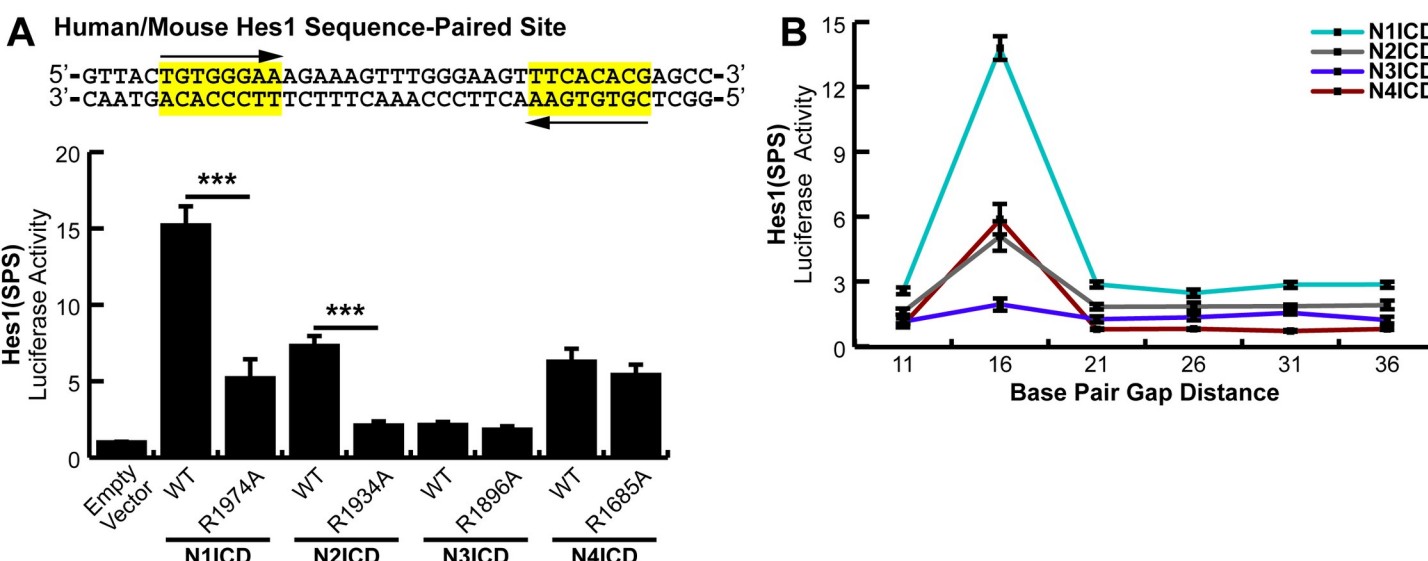

**Fig 3. Dimer dependence of the *Hes1* sequence-paired site.** (**A, Top**) Sequence information for the human and mouse isolated *Hes1* SPS element. (**A, Bottom**) HEK293T cells were transfected with the Hes1(SPS) construct and WT or dimer-incompetent versions of NICD proteins. Shown is the average +/- SE of n = 4 independent experiments. Student's t-test was performed to compare the activation between WT and mutant NICDs where * <0.05, ** <0.01, *** <0.001. (**B**) HEK293T cells were transfected with various NICDs and Hes1(SPS) constructs that had been modified to contain variable gap distances between RBPJ binding sites in 5 bp interments. Shown is the average +/- SE of 4 independent experiments.

activation (Fig 3A, Bottom) and interestingly, the isolated Hes1 SPS demonstrated approximately 5 times greater sensitivity to N1ICD compared to the full-length Hes1 promoter, most likely due to reduced background of the isolated SPS construct. Both N1ICD and N2ICD ankyrin mutants showed a significant decrease in activation compared to their wild-type counterparts. In comparison, N4ICD was again dimer independent, matching trends observed on the counterpart full-length promoter (Fig 1A). Finally, N3ICD only weakly activated the isolated Hes1 SPS. Indeed, compared to the full-length Hes1 promoter, N3ICD showed no increased activity on the isolated Hes1 SPS. Moreover, N3ICD did not demonstrate dimer dependence on the isolated Hes1 SPS but did demonstrate dimer dependence on the full-length Hes1 promoter (Fig 1A). This is potentially unsurprising since Notch3 has been shown to synergistically utilize other nearby transcription factors for its own signaling responses [9], which would be lacking in this minimalized promoter.

With a properly responding SPS-driven promoter, we again wanted to determine if long-range interactions were possible between RBPJ binding sites. We again followed the same logic employed when mutating the *Hes5* SPS and we created gap distances in steps of five nucleotides to take advantage of the periodicity of the DNA helix. We found that the wild-type, 16-nucleodtide gap, was the only distance all NICDs were capable of cooperatively binding and eliciting activation on (Fig 3B). This observation further limits the options of dimerization dependent signaling since independent NTC sites within this promoter are unlikely to cooperate over long distances by methods of kinking, looping, or untwisting the gap DNA.

## Establishment of a high activity, NICD dimer-specific reporter construct

Having found that the SPS element isolated from the Hes1 promoter functions in a dimer-dependent manner, we next sought to optimize an SPS-driven luciferase construct as a tool to specifically study the transcriptional activity of NICD dimers. To accomplish this, we constructed a synthetic SPS site that contained two high-affinity RBPJ binding sites in a head-to-head configuration separated by 16 bp. We utilized the complete TP1 consensus sequence (5′-CGTGGGAAAAT-3′) originally described by Meitinger et al., [10] thus forming the 2xTP1(SPS)-Complete construct. RBPJ shows high affinity towards this site and multiple copies of this RBPJ binding site have previously been arranged in a head-to-tail orientation to measure the activation of Notch signaling [11,30,31]. These sequences however have not previously been orientated in a head-to-head orientation to measure transcriptional activation by dimerized NICD molecules.

As shown in Fig 4A, the 2xTP1(SPS)-Complete construct successfully responded to Notch signaling. Unexpectedly however, dimer-incompetent N1ICD mutants increased activation even better than wild-type proteins indicating this promoter was not dimer-dependent. We hypothesized that since the last three nucleotides in the consensus sequence were not necessary for RBPJ-DNA interactions [10], this effectively resulted in a 22 nucleotide gap distance between RBPJ binding sites, and therefore the promoter lost its dimerization dependence. In response to this possibility, we created another SPS construct, this time with a core, essential sequence for RBPJ responsiveness (5′-GTGGGAA-3′) [10,32] separated by 16 bp. We named this new construct the 2xTP1(SPS)-Core construct. Cloning of this fragment resulted in a 'T' on the 5' side of each core sequence. This addition coincidentally matches the RBPJ consensus sequence found in Hes1/5, though this position within the consensus is variable [33]. As shown in Fig 4B, wild-type N1, N2, and N3ICD all strongly activated this promoter while the corresponding dimer-incompetent ankyrin domain mutants demonstrated significantly reduced transcriptional activity. N4ICD also activated the promoter, but remained dimerization independent, which has persisted across all SPS promoter variations tested.

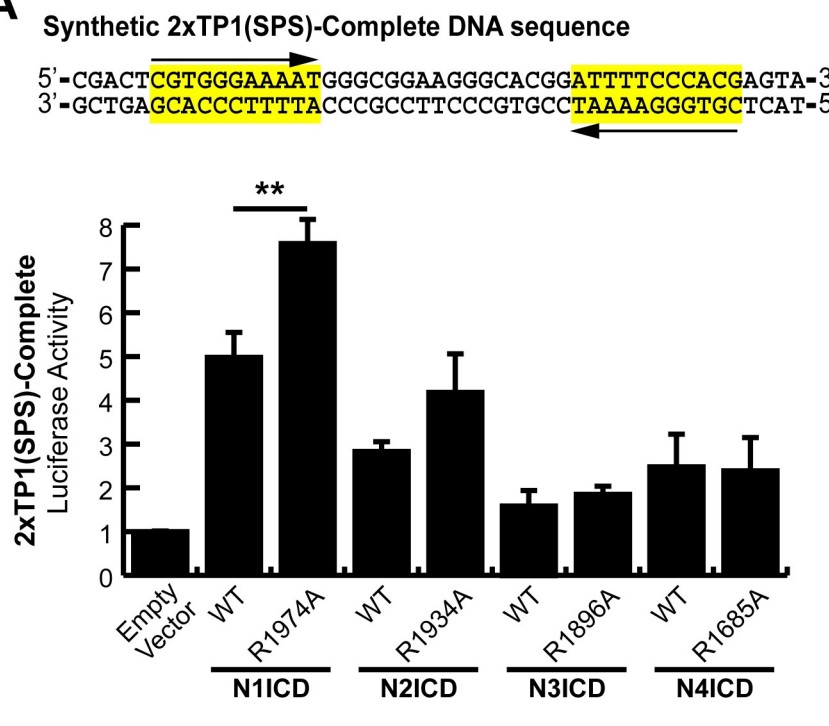

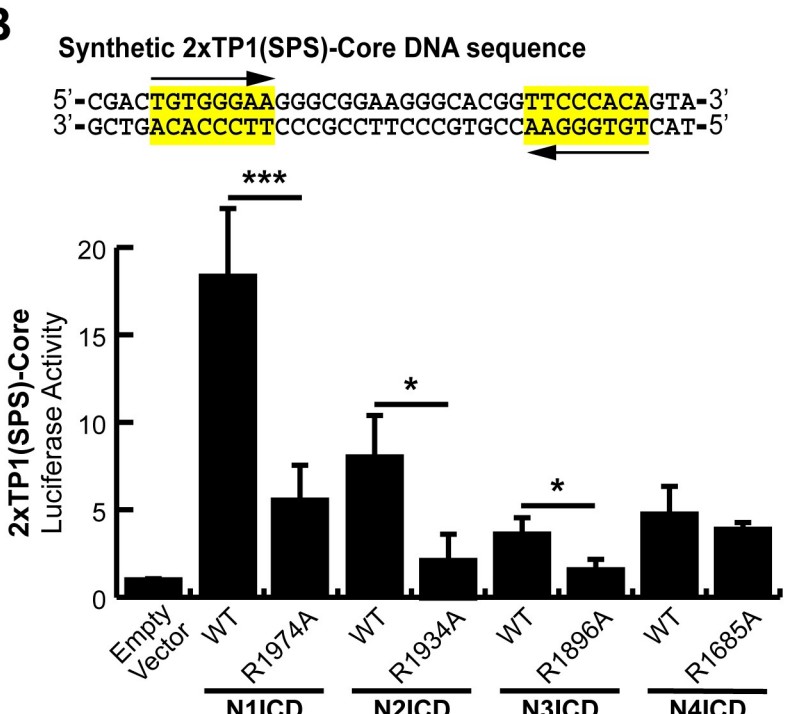

**Fig 4. An optimized Notch responsive SPS element.** (**A and B**) A synthetic SPS-driven promoter, using the (A) complete TP1 binding element or (B) core TP1 binding element, was transfected into HEK293T cells. These two constructs were activated with either wild-type (WT) or ankyrin mutated (R→A) NICDs to observe their dimer responsiveness. Shown are the average +/- SE of n = 4 independent experiments. Student's t-test was performed to compare the activation between WT and mutant NICDs where * <0.05, ** <0.01, *** <0.001.

Nonetheless, the 2xTP1(SPS)-Core construct appeared to be an optimized promoter for evaluating dimer-dependent Notch signaling.

## Non-optimal SPS sites select against transcriptional activation by NICD dimers

The results in Fig 4A revealed an interesting phenomenon wherein non-dimerizing ankyrin mutant NICDs performed better than their WT counterparts on the 2xTP1(SPS)-Complete construct which had a slightly longer gap than the 16 bp preferred by NICD molecules. This observation suggested that two RBPJ binding sites gapped slightly more or less than 16 bp within a promoter might actually suppress promoter responsiveness to NICD dimer-dependent Notch signaling and favor NICD dimer-independent Notch signaling, a phenomenon which has not been previously described. To test this hypothesis, we compared WT and ankyrin mutant N1ICD transcriptional activation from the Hes1(SPS) and 2xTP1(SPS)-Core constructs which we modified to contain head-to-head RBPJ binding sites separated by 11, 16, or 21 bp. As shown in Fig 5A and 5B, WT N1ICD strongly activated reporter transcription on the 16 bp gap promoters and was significantly weaker on the 11 bp and 21 bp gap promoters. Moreover, the ankyrin mutant N1ICD also performed as expected, showing no synergistic activity across the various gap distances. Importantly however, the ankyrin mutant N1ICD slightly outperformed the WT N1ICD on the 11 bp and 21 bp gap promoters (Fig 5A and 5B right panels). This observation suggested that head-to-head orientated RBPJ binding sites with non-optimal gap widths are more likely to be activated by non-dimerizing NICD molecules. Whether or not there is a condition that actively manipulates NICD dimerization is currently unknown.

## A restrictive spacer range dictates the signaling capabilities of NICD dimerization

Having established a highly active and NICD dimer-specific promoter, we set out to compare the promoter specificity of the various NICD molecules. To accomplish this, we established a

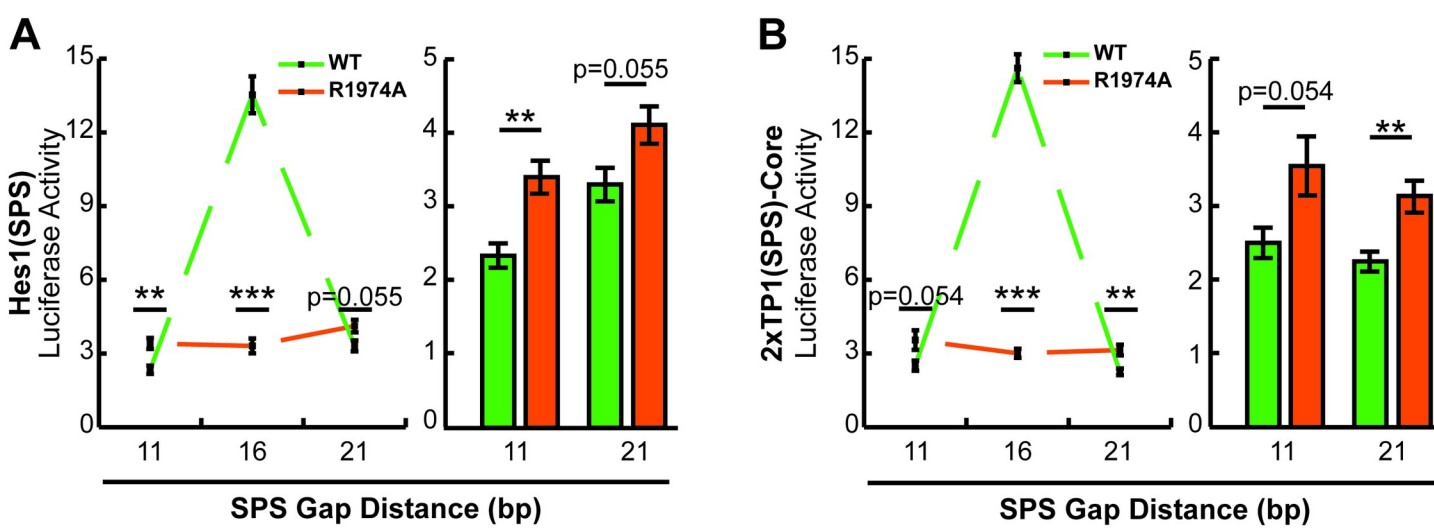

**Fig 5. Non-optimal SPS sites select against dimer-dependent NICD transcriptional activation. (A and B)** 293T cells were transfected with 11, 16, or 21 bp gap versions of the (A) Hes1(SPS) or (B) 2xTP1(SPS)-Core reporters and WT or ankyrin mutated N1ICD. Shown is the average +/- SE of n = 6 experiments. In all panels, a student's t-test was performed to compare the activation between WT and ankyrin mutated NICDs, where P-values are reported as * <0.05, ** <0.01, *** <0.001.

series of 2xTP1(SPS)-Core promoters with modified SPS gap distances ranging from 11 to 21 bp in one bp increments. Combinations of each NICD protein coupled with each spacer variation were then transfected into HEK293T cells and assayed for their transcriptional activity. Our rational for this approach was the previous finding that ChIP-Seq screening identified SPS-driven genes with various gap distances, from 11–21 base spacers [18]. Our data in Fig 3 did not support the idea that gap distances greater or less than 16 bp (in 5 bp increments) can support dimer-dependent transcription on the Hes1 promoter. Nonetheless, we wanted to use our optimized promoter to investigate the possibility that smaller variation in gap distances may be tolerated during NICD dimerization, or that different Notch family members may exhibit slightly different preferences in SPS gap width. As shown in Fig 6, all mouse NICDs have a strong preference for the SPS sites with 16 bp gaps, with little room for variation. For N1ICD, there was some flexibility observed with 15 and 17 bp spacers, though anything outside of this range did not demonstrate cooperative signaling (Fig 6A). In contrast, N2ICD showed even more specificity, where there was only slight cooperativity at the 15 bp gap, and none observed at 17 (Fig 6B). N3ICD showed cooperativity at 16 bases (Fig 6C), and N4ICD at 16 and 17 bases (Fig 6D), however it's difficult to make a judgement call about their flexibility due to their low activation and their apparent dimerization independent signaling. Also, it's worth noting is that N4ICD preferred a 16 bp gap suggesting that N4ICD was functioning as a dimer which is in conflict with our earlier observations that N4ICD functioned dimer-independently. Moreover, N4ICD demonstrated slight inhibition on promoters with 11 or 14 bp gaps, which was similarly observed in the Hes5 (Fig 2C) and Hes1 (Fig 3B) SPS promoter constructs. This might indicate that N4ICD proteins generally act as transcription inhibitors on monomeric NTC sites, or perhaps occupy RBPJ binding sites and prevent other NICDs from binding, though we did not explore these possibilities. Further we also observed a slight increase in activation of the 12 or 13 bp gap constructs for all four NICD proteins, which could be attributed to some form of cooperativity, though we did not explore this thread any further. Finally, as with the Hes5(SPS) and Hes1(SPS), we extended the gap distances of the 2xTP1 (SPS)-Core out to 36 nucleotides and again observe little to no long-range cooperativity.

## Discussion

Notch signaling is an important cellular communication mechanism that is required for multicellular organisms. Ongoing research continues to reveal how Notch functions and how Notch signaling is integrated into many facets of cell biology. Despite our growing understanding of Notch function, however, there are basic questions about Notch signaling that remains to be addressed. In this study, we sought to address some of these basic questions about Notch signaling that have been overlooked in the quest to dig into the deeper questions of Notch function. In particular, we felt that a head-to-head transcriptional comparison of the full-length Notch NICD molecules on a variety of promoter elements should be performed. Most of what we know about Notch dimerization, including how the proteins interact and on which promoters, comes from studies with Notch1. While the four mammalian Notch proteins are mostly homologous within the N-terminal and ankyrin domains, there are substantial differences within the C-terminal regions. Notably, this domain is absent in much of the field's previous characterization work. Indeed, molecular modeling [15] and crystallization [16] were performed with just the ankyrin domain, and these studies laid the original groundwork for the NTC's spatial conformations, interacting amino acids, and DNA preferences. Similarly, further *in vitro* work with EMSAs and FRET assays supported NTC cooperative loading and SPS gap preferences, though these were performed using N1ICDs with just the N-terminal RAM and ankyrin domains [15,16,19]. Importantly, the C-terminus has known transcriptional

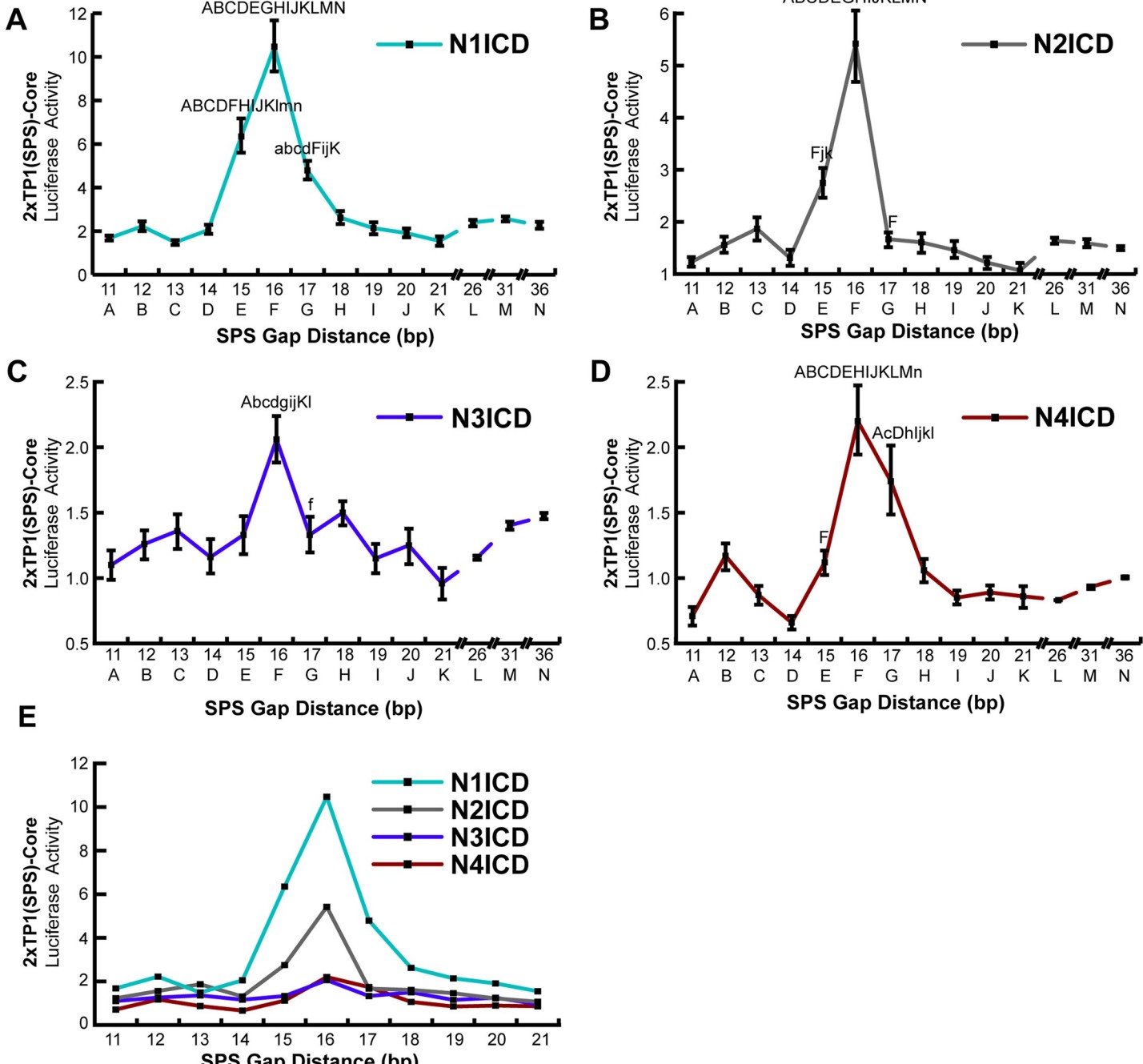

**Fig 6. Notch homodimers do not tolerate alternative gap distances.** (**A-D**) HEK293T cells were co-transfected with various NICD expression plasmids and 2xTP1 (SPS)-Core promoter with varying SPS gap lengths from 11–21 bp, in one bp increments, or 21–36 bp, in 5 bp increments. For all luciferase studies, an empty expression vector was used to equate the amount of DNA transfected across samples. Fold differences are represented as the NICD-induced activity compared to the basal activity levels. Shown are the average +/- SE of n = 5 independent experiments. (**E**) Overlay of data in panels A-D to emphasize different transcriptional strengths of the various NICDs. Statistical analysis was performed by using a one-way analysis of variance (ANOVA), followed by a Tukey-Kramer's HSD post-hoc test. Only points that were statistically different from other points are indicated in each graph where capital letters indicate P values < 0.001, whereas small case letters indicate P < 0.05 on a point-by-point basis.

effects [34], regulatory capacity [35], and contains a multimerization site [36]. Therefore, it was important to ensure 1) the presence of these domains don't interrupt dimerization responses in a cellular context, and 2) if the lessons learned by studying Notch1 can be broadly applied to the other mammalian Notch proteins.

To address these concerns, we first revisited full-length, Notch-responsive promoters, which contain canonical sequence-paired sites to drive transcription. While these promoters did require NICD dimerization for full activation and therefore allowed for preliminary conclusions, our data collection suffered from low activation levels (Hes1) or high experimental variability (Hes5). This is a logical byproduct from using full-length promoters since these genes are regulated through mechanisms beyond just Notch signaling and overexpression of constitutively active NICDs would have wide, overarching, signaling outcomes. For example, as in the case of Hes1, this gene is understood to be self-regulated, resulting in its oscillatory nature [37]. For our purposes, this implies that endogenous Hes1 may be activated by NICD overexpression and circles back to our Hes1-luciferase promoter to negatively-regulate it, resulting in its poor overall activation. For reasons like this, we sought to create minimalized promoters to more specifically monitor dimerization-induced signaling.

Since synthetic promoters with tandemly arranged RBPJ sites had been previously used to monitor Notch activity (e.g. 4xCSL-Luc, [23]), we reasoned that the SPS alone should also be sufficient for Notch activation. Therefore, we isolated the known sequence-paired sites out of Hes1 and Hes5 and created the Hes1/5(SPS)-luciferase vectors. Since the reverse sites within Hes1 and 5 do not perfectly match the canonical RBPJ binding site, we also crafted the synthetic 2xTP1(SPS)-Core promoter to make a "perfect" SPS element. Upon testing we found that the Hes1(SPS) and the 2xTP1(SPS)-Core promoters operated as expected in that they were robustly activated by N1ICD and inefficiently activated by dimer-incompetent NICD ankyrin mutants. Surprisingly however, our results did not indicate that the isolated Hes5 SPS site functions in a dimer-dependent manner, although the full-length Hes5 promoter was dimer-dependent. We return to this point later in this discussion.

Having established two reporters with high activity and specificity for dimerized NICD signaling we were able to specifically ask questions about dimerization of full-length NICD proteins in cells. We found that full-length N1ICD and N2ICD activated sequence-paired sites in a dimerization-dependent mechanism. In contrast, N3ICD's activation of these minimal promoters was poor, which may be partially attributed to its lack of a TAD domain in its C-terminal [34] or usage of other transcription factors for its activation [9]. It also differed in its dimer-dependence; where it was not dimer-dependent on the Hes1(SPS) but was on the 2xTP1(SPS)-Core. This implies there is some difference located within these two minimalized constructs that can affect dimerization outcomes. The primary source of variation can be found within the 16 bp gap between RBPJ binding sites. The original crystallography studies found that the Hes1 gap distance DNA must go through substantial untwisting to bring the ankyrin domains into contact with each other for dimerization [16]. Gap composition of the 2xTP1(SPS)-Core construct has even higher G/C content than Hes1's, so while N1ICD and N2ICD can utilize variable gap compositions, perhaps N3ICD utilizes a more inflexible gap sequence. Whether or not this would enable differential transcriptional activity of Notch remains to be tested. In contrast, however, N4ICD did not appear to have dimer-dependent activation on the Hes1(SPS) or 2xTP1(SPS)-Core constructs despite still displaying a preference for a 16 bp gap between RBPJ binding sites on these promoters. Based on these results, we believe that the single R1685A mutation (which ablates dimerization of the other NICDs) on N4ICD was insufficient to abolish its dimerization activity suggesting that perhaps N4ICD has an alternative dimerization interface. Indeed, crystallography work on human N1ICD homodimers indicated two other amino acids (K1945 and E1949) beyond the R1984 that are

involved in ankyrin-mediated dimerization [16]. Interestingly the N4ICD is the only NICD with an "R" located at the K1945 site, suggesting this position may be especially important for N4ICD dimerization (S2 Fig).

In addition to comparing the activation parameters of the various NICDs on these optimized promoters, we were also curious about the standard 16-nucleotide gap distance established by the N1ICD containing tripartite complex. Two hypotheses presented themselves to us. First, given the contortion of DNA evident in the crystalized N1ICD trimeric complex, we hypothesized that RBPJ sites outside of the optimal 16 bp gap might be utilized through further contortions of DNA between RBPJ binding sites. Second, we further hypothesized that given the variable sequences and sizes of the NICD proteins, different NICDs might have preferences for slightly longer or shorter gap distances between RBPJ binding sites. To test the first hypothesis, we created SPS sites with variable gap distances in 5 bp increments to take advantage of the periodicity of helical DNA. We found that none of these long-range alternative gap distances, for any of the NICDs, had significantly enhanced signaling above monomeric signaling. This indicated that long-range interactions between NTC complexes are unlikely to occur. Our analysis however only extended to measuring 36 bp gaps and therefore, longer-range interactions between RBPJ binding sites cannot be ruled out. To address the second hypothesis (that different sized NICDs might have subtle differences in SPS preference), we compared NICD activity on SPS sites with 11–21 bp gap distances in one bp increments. Similar to previous results [16,19,38], we found that all full-length NICDs also prefer a 16 bp gap distance, with little room for variation. Based on this result, we conclude that the C-terminal regions of NICDs do not appear to change the SPS dimensions preferred by NICDs nor impact NICD dimerization. It should also be stated, however, that deletion of the N1ICD C-terminal tail (ΔS2184) which contains the trans-activation domain (TAD) significantly reduced transcriptional activity (S3 Fig). Minor differences in the NICDs however were observed. For example, N2ICD appeared slightly more restrictive when choosing SPS sites than Notch1, since N2ICD activated 15/16 bp spacers but not the 17 bp gap. This poses an interesting possibility where N1ICD might dimer-dependently activate *CUL1* or *TXN2*, which have SPSs with a 17-nucleotide gap distance [16,18], whereas binding of N2ICD on this promoter would only cause monomeric levels of transcription, though this idea was left untested. While N3ICD and N4ICD had questionable dimerization activities, they did seem to have cooperative binding and higher activity on the 16/17 bp gap SPSs. Making firm conclusions about their preferences is difficult however due to their low overall activity and high variability. Together, these experiments highlight the stringent requirements for dimerization-induced transcription.

Our results have also shed some light on the nature of NICD dimerization. First, there has been some debate as to the order of events leading up to NICD activated transcription. It is currently unclear whether individual NICD molecules first bind to RBPJ/DNA then form dimers, or alternatively if two NICDs first form a dimer, then bind to RBPJ/DNA. In Figs 4 and 5, we demonstrated that dimer-incompetent N1ICD performed slightly better than WT N1ICD on SPS elements with non-optimal 11 or 21 bp gaps. This suggests that the attempt to dimerize may impede NICD molecules from binding to these sites and might be a clue as to the mechanics of NICD function. Our result indicates that non-optimal SPS elements discourage NICD dimer formation and we believe this is evidence that NICD molecules are pre-forming dimers before binding to RBPJ/DNA. Whether or not this has an actual impact on how promoters with non-optimal SPS sites are utilized by Notch signaling remains to be seen but as shown by Castel et al., [18], several Notch responsive promoters with non-optimal SPS elements have been identified. Another outstanding question about NICD dimerization is whether or not NICD molecules can engage in heterodimerization. Given the conservation of sequence in the ankyrin domains and the importance of ankyrin domains for NICD

dimerization it has been hypothesized that NICD heterodimers may exist. While further research on this topic is certainly warranted, our results in Fig 6 showing similar SPS element preferences and in Fig 4 showing ChIP recovery of N1/N4 complexes suggests that heterodimers between N1ICD and N4ICD can form in transfected 293T cells and therefore possibly under more physiological conditions. Here again, the biological implications of this observation are unknown but given the strong differences in transcriptional activation between N1ICD and N4ICD, we hypothesize that N1/N4ICD heterodimers would have intermediate activity compared to N1ICD or N4ICD homodimers. Thus, heterodimerization of NICD molecules may offer a new mechanism to regulate outputs from Notch signaling.

Finally, our minimized Hes5(SPS) promoter was not sufficient to elicit dimerization-dependent activation. The Hes5 promoter does not contain a canonical sequence-paired site and instead has been described as 'cryptic' [16], with a standard forward RBPJ binding site but an abnormal reverse site. When arranged as a palindrome in a SPS, this abnormal reverse site does not support dimerization, yet it facilitated dimerization when paired with a strong RBPJ binding site [19]. Work with EMSAs of this cryptic SPS showed that N1ICD homodimers can form, and dimerization-dependent activation through this site was supported through luciferase assays [16], but distinctly, these luciferase assays were still performed with the full-length promoter. In our analyses we isolated out this SPS, which should be sufficient for dimerization, yet this construct did not demonstrate dimerization-dependent activation.

This inconsistency poses two thought-provoking problems. First, and as originally described by Severson et al. [19], if cryptic sequence-paired sites are capable of forming NICD dimers, then searching for SPSs by 'sequence-gazing' becomes far more difficult. For example, previous ChIP work isolated out RBPJ-bound DNA targets, and these sequences were screened for nearby RBPJ binding motifs located in tandem [18]. The issue here lies in the partner sequence, wherein any non-conforming RBPJ sequences would be missed through a simple screening approach. To further identify other cryptic sequence-paired sites, like those in Hes5 or $pT\alpha$ [16,17], we propose that a logical course of action is to perform ChIP-Seq on a double-selected pool of DNA that specifically identifies two dimerizing NICD partners. The second problem concerns the activation of Hes5 through its SPS. While it appears that NTC complexes dimerize on this promoter segment *in vitro*, do they still form in living cells and if so, what's the missing link for transcriptional activation? Other than the cryptic RBPJ binding site, the Hes5(SPS) construct is nearly identical to the Hes1(SPS) and 2xTP1(SPS)-Core constructs which respond as expected. Since the full-length Hes5 promoter is apparently dimer-responsive and the other minimalized SPS constructs are sufficient for activation, we predict that there are other promoter elements involved which enable these low-affinity dimers to form and signal in a cellular context. Comparing the promoters and enhancers of multiple, cryptic, SPS genes may identify other sequence motifs in common and identify signaling or regulatory pathways involved.

In conclusion, since much of the work on NICD dimerization has been performed studying C-terminally truncated N1ICD, we felt it was important to 1.) examine full-length NICD molecules to insure the C-terminal domain does not affect NICD promoter preference, and 2.) compare promoter preferences of the other NICD molecules, which have been largely overlooked. In so doing, we confirmed that while the C-terminal domain of the various NICDs has a transcriptional role, this domain does not appear to play a role in promoter preference. In addition, we also found that all the mammalian NICDs have remarkably similar SPS gap length preferences with only minor (+/- one bp) flexibility. Overall, these results both support previous work but also help fill in missing gaps in our understanding of Notch transcriptional activity.

## Supporting information

**S1 Fig. Identification of low and high affinity SPS sites in Hes1 and Hes5 promoters.** DNA sequences are as reported for Addgene products Hes1-luciferase (#41723) and Hes5-luciferase (41724) vectors. Possible high and low affinity sites (as described in material and methods) are indicated by forward or reverse arrows. High affinity sites confirmed to serve as RBPj binding sites are indicated with red shading. Nucleotides that diverge from the core TP1 element as originally defined (C/tGTGGGAA) are indicated by bolded letters.
(TIF)

**S2 Fig. Sequence comparison of human N1ICD and mouse NICD dimerization domains.** Amino acids previously identified as important for dimerization of N1ICD ankyrin domains are indicated [16]. Amino acid numbers are based on equivalent positions in human N1ICD. R1984A equivalents were used throughout this work to attempt to make generate dimer-incompetent NICD variants.
(TIF)

**S3 Fig. N1ICD C-terminal does not impact NICD dimerization.** HEK293T cells were transfected with 11, 16, or 21 bp versions of Hes1(SPS) or 2xTP1(SPS)-Core luciferase reporters and either full-length N1ICD or ΔS2184 N1ICD (which deletes residues C-terminal of the N1ICD ankyrin domain). Shown is the average +/- SE of four independent experiments.
(TIF)

**S1 Table. Plasmid and sequence information for the NICD-activated sequence-paired sites.**
(DOCX)

**S1 Raw images.**
(PDF)

## Author Contributions

**Conceptualization:** Jacob J. Crow.

**Formal analysis:** Jacob J. Crow.

**Investigation:** Jacob J. Crow, Allan R. Albig.

**Project administration:** Allan R. Albig.

**Supervision:** Allan R. Albig.

**Writing – original draft:** Jacob J. Crow.

**Writing – review & editing:** Jacob J. Crow, Allan R. Albig.

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
