## [Decision Letter · Decision Letter 0]

5 Jul 2020

PONE-D-20-14835

Notch family members follow stringent requirements for intracellular domain dimerization at sequence-paired sites

PLOS ONE

Dear Dr. Albig,

Thank you for submitting your manuscript to PLOS ONE. After careful consideration, we feel that it has merit but does not fully meet PLOS ONE’s publication criteria as it currently stands. Therefore, we invite you to submit a revised version of the manuscript that addresses the points raised during the review process.

I recommend that you pay particular attention and address the points raised by reviewer 1. 

We look forward to receiving your revised manuscript.

Kind regards,

George Mosialos

Academic Editor

PLOS ONE

Journal Requirements:

2.PLOS ONE now requires that authors provide the original uncropped and unadjusted images underlying all blot or gel results reported in a submission’s figures or Supporting Information files. This policy and the journal’s other requirements for blot/gel reporting and figure preparation are described in detail at https://journals.plos.org/plosone/s/figures#loc-blot-and-gel-reporting-requirements and https://journals.plos.org/plosone/s/figures#loc-preparing-figures-from-image-files. When you submit your revised manuscript, please ensure that your figures adhere fully to these guidelines and provide the original underlying images for all blot or gel data reported in your submission. See the following link for instructions on providing the original image data: https://journals.plos.org/plosone/s/figures#loc-original-images-for-blots-and-gels.

Reviewers' comments:

Reviewer's Responses to Questions

**Comments to the Author**

1. Is the manuscript technically sound, and do the data support the conclusions?

Reviewer #1: Partly

Reviewer #2: Yes

2. Has the statistical analysis been performed appropriately and rigorously? 

Reviewer #1: Yes

Reviewer #2: Yes

3. Have the authors made all data underlying the findings in their manuscript fully available?

Reviewer #1: Yes

Reviewer #2: Yes

4. Is the manuscript presented in an intelligible fashion and written in standard English?

Reviewer #1: Yes

Reviewer #2: Yes

5. Review Comments to the Author

Reviewer #1: Review of: “Notch family members follow stringent requirements for intracellular domain dimerization at sequence paired sites.”

Summary: The focus of this paper is to compare the transcriptional activity of Notch1, 2, 3, and 4 wild type and mutants containing a specific amino acid change in an analogous residue of each molecule that was previously shown to disrupt NICD-1 dimer formation on SPS sites. The general strategy is to use cell culture and over-expression assays coupled with an array of different luciferase vectors containing Rbpj/CSL binding sites in different orientations and with different spacers to assess transcriptional changes. The authors further use Western Blot analysis to compare NICD protein levels, and perform a ChIP-PCR study to assess for protein binding to DNA.

The authors make several conclusions based on their data including, A) That all Notch ICD molecules can form dimers on the SPS sites; B) Each Notch dimer requires similar spacing and site orientation as that published for Notch1. C) The authors claim to have found a “mechanical difference between canonical and cryptic SPSs, leading to differences in their dimerization-induced regulation.” D) Notch1 and Notch4 can form heterodimers on SPS sites. Overall, the authors have provided evidence for some of these conclusions, but not all. Hence, I have the following concerns in regards to the experimental conclusions/interpretations and the experimental design of the study.

1) An assumption made in the paper is that the same mutation that disrupts Notch1-ICD dimer formation will also disrupt N2, N3, and N4 dimer formation. Past publications used assays like EMSAs or FRET assays to show that Notch1-ICD forms dimers, but the authors provided no direct evidence that the mutations tested in the luciferase assays actually abolishes dimer formation for the other Notch molecules. Hence, any change in transcriptional activity is assumed to be caused by a decrease in dimer formation without any evidence it actually does cause that change. They need to provide such evidence or they need to dramatically change the way they write the paper about whether these mutations actually impact cooperative DNA binding.

2) A better description of the plasmid design of the synthetic luciferase reporters is needed. In particular, I did not understand how these plasmids work given the authors wrote the following: “We isolated the mouse and human Hes5 SPS elements containing two head-to-head orientated RBPJ binding sites separated by a 16-nucleotide gap, and the 4 to 5 surrounding nucleotides (Figure 2A) as previously described and cloned these sequences into promoterless luciferase reporter vectors.” The authors similarly stated that they used a promoterless pGL3-basic vector in the methods. If the luciferase vector does not have a promoter and the synthetic insert only has an SPS site – then how does RNA polymerase initiate transcription? Is the assumption here that the constructs initiate transcription in the complete absence of a promoter or does it contain a minimal promoter that is insufficient to activate gene expression on its own? If it completely lacks a promoter – the authors should provide a clear rationale for why it provides a good model to use to study Notch-dependent transcription, as all Notch target genes presumably have a promoter. Given this discrepancy, the relevance of the results can be called into question.

3) The authors claim in the abstract that “each family member is capable of dimerization-induced signaling”, but the authors provide no evidence for Notch4. The only evidence the authors show is that N4 induces optimal induction of a 2xTP1(SPS)core construct with a 16bp spacer in Figure 6. But the authors used that same luciferase reporter to test the N4 wild type and mutant protein and found no significant change in activation using the mutant N4 molecule in Figure 4B. Hence, no N4 mutant protein was shown to significantly change transcription via SPS sites or any other site. Hence, the authors do not have the data to support this claim.

4) The authors also did double-selection ChIP-PCR experiments in Figure 4 to make the following claim: “Our analysis determined that both Notch1 and Notch4 homodimers cooperate on the 2xTP1(SPS)-Core construct and are therefore likely actively signaling as a duplex (Figure 4C). Importantly, we also observed Notch1-Notch4 heterodimers, though at a lesser abundance. This implies that two different Notch proteins can come together at the same promoter site and interact, possibly modulating the other’s transcriptional output.” I have several concerns about the interpretation of this experiment. First, the authors stated that these homodimers bind in a “cooperative” manner. However, no proper positive or negative controls are included to demonstrate cooperativity. For example, the ChIP-PCR experiments should also be done on a region containing two high affinity binding sites that do not mediate cooperative binding (i.e. 21bp spacer) and show that there is a dramatic decrease in binding – which would be consistent with cooperativity. Second, the authors need a negative control by testing a reporter with both binding sites mutated. Third, it is standard practice to quantify this data using a fold enrichment over input chromatin. Fourth, the authors should also test the dimer mutations – especially to make the claim that N4/N1 heterodimers form – given that the authors have not convincingly shown that N4 even forms homodimers (see point 3 above).

5) The authors made the following statement in the results section: “The results in figure 4A revealed an interesting phenomenon wherein non-dimerizing ankyrin mutant NICDs performed better than their WT counterparts on the 2xTP1(SPS)-Complete construct which had a slightly longer gap than the 16 bp preferred by NICD molecules. This observation suggested that two RBPJ binding sites gapped slightly more or less than 16 bp within a promoter might actually suppress promoter responsiveness to NICD dimer-dependent Notch signaling and favor NICD dimer-independent Notch signaling, a phenomenon which has not been previously described.” However the change was only from ~5-fold for the wild type protein to ~7.5-fold for the RA mutation. Couldn’t such a small relative change in luciferase activity between the wild type vs RA mutant protein actually be explained by differences in the levels of the NICD wild type and RA proteins? In looking at the Western blots in Figure 1C, it appears that the NICD-RA proteins are consistently higher than the Wild type protein – although no quantitation is provided. This increase in protein levels could explain why the RA protein is better on the 11bp and 21bp spacer luciferase constructs. Note, the increase in levels of the RA protein relative to wild type could be due to increased protein stability as has been recently suggested in Drosophila by Kuang et al eLife 2020. However, another recent paper (Kobia et al 2020 on bioarchives) found that the N2RA mutation did not dramatically alter N2ICD levels – and thus such changes in levels/stability could be cell type specific.

6) The authors should better describe how they selected the sequences used in the different spacers. Were they selected to exclude additional TF binding sites? Some previously described SPS sequences were found to often contain a common hexamer sequence (typically something like GAAAGT, which is found in Hes1, see Posakony papers). Did they purposely avoid or include such sequences in their constructs? Did they scan the sequences to exclude the possibility of making additional low affinity Rbpj sites?

7) Minor comment: Figure 6 needs statistical tests.

Reviewer #2: In this interesting study, Crow and Albig dissect the requirements and activity for the dimerization of all 4 different NOTCH proteins using reporter assays in 293T cells. Authors show that all members can form dimers and can even form heterodimers, while the sequence GAP for dimer binding-sequences is mostly limited to the originally described 16bp gap. I think this study will positively contribute to the understanding of the NOTCH field.

I only have very minor comments:

1. Authors shouldn't refer to "dimerization null constructs", but rather "dimerization incompetent constructs" throughout the text.

2. In Fig 1D, it would be good if authors could exactly specify the high/low affinity DNA sequences identified in a Supplementary Fig.

3. In the discussion, authors hypothesize that R1945 might be relevant in N4ID for dimerization. Can authors check this? This should be fairly easy to do using their reporter system, upon site-directed mutagenesis for that position.

4. Please correct grammar in the Discussion in the sentence "This indicated

that long-range interactions between NTC complexes are unlikely TO occur"

6. PLOS authors have the option to publish the peer review history of their article (what does this mean?). If published, this will include your full peer review and any attached files.

Reviewer #1: No

Reviewer #2: No

---

## [Author Response · Author response to Decision Letter 0]

20 Aug 2020

Thank you for serving as editor for our submission to PLOS One. We thank the reviewers for their thoughtful review of our manuscript and we were honored by the overall enthusiasm for our work. We have addressed the reviewers concerns on a point-by-point basis below. In each case, we repeat the reviewers concern and follow with our response below each point.

Reviewer #1: 

1) An assumption made in the paper is that the same mutation that disrupts Notch1-ICD dimer formation will also disrupt N2, N3, and N4 dimer formation. Past publications used assays like EMSAs or FRET assays to show that Notch1-ICD forms dimers, but the authors provided no direct evidence that the mutations tested in the luciferase assays actually abolishes dimer formation for the other Notch molecules. Hence, any change in transcriptional activity is assumed to be caused by a decrease in dimer formation without any evidence it actually does cause that change. They need to provide such evidence or they need to dramatically change the way they write the paper about whether these mutations actually impact cooperative DNA binding.

- The reviewer is concerned that the amino acid mutation that renders N1ICD dimer incompetent, may not also cause the other NICDs to be dimer incompetent. The reviewer is also concerned that reduced transcriptional activity of NICD mutants might be due to altered protein expression or stability. This is a fair concern and one that we would have loved to address with EMSA assay (as suggested). However, despite our best attempts, we have been unable to make the NICD EMSA assay work in our hands. As an alternative, we have resorted to our luciferase promoter assay to address the reviewer's concerns. In our new data, we compared the transcriptional activity of both WT and mutated NICD molecules on a dimer-independent 4XTP1 luciferase construct. We rationalized that if WT and mutant NICD performed equivalently on this 4XTP1 that this would eliminate any concerns about protein expression, stability, or interactions with CSL and leave the only explanation for reduced activity on dimer-dependent promoters that the mutation blocks dimerization activity. Our new data in figure 1D shows that all the WT and mutant NICD molecules in fact do have equivalent activities on this dimer-independent construct suggesting that changes in transcriptional activity are not related to protein expression or stability. As has been noted elsewhere in this rebuttal letter, N4ICD is the exception since it does not appear to function in a dimer dependent manner based on mutation of the ankyrin domain. That said, our data does demonstrate what appears to be dimerization activity by ChIP and by the preference of N4ICD for a 16 bp template. Based on these observations, we do feel that N4ICD engages in dimerization, but that the amino acid mutated in our present experiments "missed the mark". Instead, future research is going to be focused on identifying the critical amino acids in N4ICD that are critical for dimerization activities.

2) A better description of the plasmid design of the synthetic luciferase reporters is needed. In particular, I did not understand how these plasmids work given the authors wrote the following: "We isolated the mouse and human Hes5 SPS elements containing two head-to-head orientated RBPJ binding sites separated by a 16-nucleotide gap, and the 4 to 5 surrounding nucleotides (Figure 2A) as previously described and cloned these sequences into promoterless luciferase reporter vectors." The authors similarly stated that they used a promoterless pGL3-basic vector in the methods. If the luciferase vector does not have a promoter and the synthetic insert only has an SPS site - then how does RNA polymerase initiate transcription? Is the assumption here that the constructs initiate transcription in the complete absence of a promoter or does it contain a minimal promoter that is insufficient to activate gene expression on its own? If it completely lacks a promoter - the authors should provide a clear rationale for why it provides a good model to use to study Notch-dependent transcription, as all Notch target genes presumably have a promoter. Given this discrepancy, the relevance of the results can be called into question.

- The reviewer is absolutely correct. The term "promoterless" is a sort of shorthand commonly used to describe a plasmid that has a minimal promoter, incapable of initiating transcription without additional enhancer elements (such as the SPS sites) used in our paper. We have clarified this throughout the manuscript.

3) The authors claim in the abstract that "each family member is capable of dimerization-induced signaling", but the authors provide no evidence for Notch4. The only evidence the authors show is that N4 induces optimal induction of a 2xTP1(SPS)core construct with a 16bp spacer in Figure 6. But the authors used that same luciferase reporter to test the N4 wild type and mutant protein and found no significant change in activation using the mutant N4 molecule in Figure 4B. Hence, no N4 mutant protein was shown to significantly change transcription via SPS sites or any other site. Hence, the authors do not have the data to support this claim.

-We agree with the reviewer that our evidence for N4ICD dimerization is very minimal. Unfortunately, I believe that the amino acid mutation that seems to "break" dimerization in the other NICD molecules does not affect N4ICD in the same way. We have enhanced our previous figure 1 (now figure S2) to show an expanded sequence alignment that provides a better understanding into how the assumed dimerization domain is fairly different in N4ICD compared to the other NICDs. Regardless, we cannot say for sure that N4ICD does (or does not) dimerize. As the reviewer notes, our only evidence is a preference for the 16bp gap and the CHiP data in panel 4C wherein we detect N4ICD dimers by Co-IP and PCR. We attempted to emphasize this throughout the text but overlooked this comment in the abstract. This has been amended. This comment is also similar to one of reviewer two's comments. We are currently performing experiments to determine which amino acids in N4ICD are necessary for dimerization.

4) The authors also did double-selection ChIP-PCR experiments in Figure 4 to make the following claim: "Our analysis determined that both Notch1 and Notch4 homodimers cooperate on the 2xTP1(SPS)-Core construct and are therefore likely actively signaling as a duplex (Figure 4C). Importantly, we also observed Notch1-Notch4 heterodimers, though at a lesser abundance. This implies that two different Notch proteins can come together at the same promoter site and interact, possibly modulating the other's transcriptional output." I have several concerns about the interpretation of this experiment. 

First, the authors stated that these homodimers bind in a "cooperative" manner. However, no proper positive or negative controls are included to demonstrate cooperativity. For example, the ChIP-PCR experiments should also be done on a region containing two high affinity binding sites that do not mediate cooperative binding (i.e. 21bp spacer) and show that there is a dramatic decrease in binding - which would be consistent with cooperativity. 

-Thank you, reviewer, for pointing this out. We were mistaken to suggest that we were interested in demonstrating cooperativity in this experiment. Instead, we were only interested in observing NICD dimerization. We have extensively rewritten this section in the revised manuscript. Demonstrating cooperativity would be extremely challenging by this approach since as we understand the assembly of the Notch tripartite complex, the synergy of the 16bp template is not associated with NICD/RBPj binding to DNA, but rather to the ability to recruit RNA polymerase. Therefore, since the 21 bp promoter can still theoretically bind to two differently tagged (but undimerized) NICD molecules, we would probably still detect ChIP on the 21 bp promoter. A better readout for cooperative function is in fact strong transcriptional activity on the 16bp template which is demonstrated in figure 6.

Second, the authors need a negative control by testing a reporter with both binding sites mutated. 

-We also originally thought about using mutated SPS sites as a negative control in this experiment. However given the sensitivity of PCR as a readout, we were concerned that any residual binding (even on mutated sites) would be detected in our PCR analysis. Since we were also not attempting a quantitative approach to our analysis, this would have been difficult to control for. In the end, we feel that the two negative controls we did use are excellent. The most basic negative control we used was just the luciferase plasmid transfected with no NICD. The more robust negative control we used was the sample transfected with HA-tagged NICD and lacking Flag-tagged NICD, but still subjected to the two-step Co-IP procedure. We felt (and still do) that the absence of PCR signal in this more robust control provides very good evidence for the specificity and lack of background in our experiment.

Third, it is standard practice to quantify this data using a fold enrichment over input chromatin. 

-It was never our intention (at this point anyway) to use this ChIP assay for anything more than a demonstration of another technique to detect dimerized NICD molecules on DNA. 

Fourth, the authors should also test the dimer mutations - especially to make the claim that N4/N1 heterodimers form - given that the authors have not convincingly shown that N4 even forms homodimers (see point 3 above).

-On the surface, this sounds like an excellent control for our experiment however it is necessary to remember that any two NICD molecules might be able to bind to the DNA regardless if they are dimer capable or incapable. Therefore, this experiment would probably give false positives, suggesting dimerization even though it was not happening. As stated above, we really feel the controls we chose to use in this study are a good compromise to show specificity of the results and to avoid the potentially sticky false positives we sought to avoid.

5) The authors made the following statement in the results section: "The results in figure 4A revealed an interesting phenomenon wherein non-dimerizing ankyrin mutant NICDs performed better than their WT counterparts on the 2xTP1(SPS)-Complete construct which had a slightly longer gap than the 16 bp preferred by NICD molecules. This observation suggested that two RBPJ binding sites gapped slightly more or less than 16 bp within a promoter might actually suppress promoter responsiveness to NICD dimer-dependent Notch signaling and favor NICD dimer-independent Notch signaling, a phenomenon which has not been previously described." However the change was only from ~5-fold for the wild type protein to ~7.5-fold for the RA mutation. Couldn't such a small relative change in luciferase activity between the wild type vs RA mutant protein actually be explained by differences in the levels of the NICD wild type and RA proteins? In looking at the Western blots in Figure 1C, it appears that the NICD-RA proteins are consistently higher than the Wild type protein - although no quantitation is provided. This increase in protein levels could explain why the RA protein is better on the 11bp and 21bp spacer luciferase constructs. Note, the increase in levels of the RA protein relative to wild type could be due to increased protein stability as has been recently suggested in Drosophila by Kuang et al eLife 2020. However, another recent paper (Kobia et al 2020 on bioarchives) found that the N2RA mutation did not dramatically alter N2ICD levels - and thus such changes in levels/stability could be cell type specific.

-The reviewer is correct that this small change could be caused by properties of the mutant N1ICD other than dimerization potential. This concern however is addressed in the revised figure 1D wherein we compare WT and ankyrin mutant NICD transcriptional activity on dimer independent template 4xTP1. We found no significant difference between WT and mutant on this promoter suggesting that any difference between WT and mutant NICDs is based on dimerization potential and the synergistic transcriptional activity associated with NICD dimerization.

6) The authors should better describe how they selected the sequences used in the different spacers. Were they selected to exclude additional TF binding sites? Some previously described SPS sequences were found to often contain a common hexamer sequence (typically something like GAAAGT, which is found in Hes1, see Posakony papers). Did they purposely avoid or include such sequences in their constructs? Did they scan the sequences to exclude the possibility of making additional low affinity Rbpj sites?

-We have revised the section in the material and methods that describes plasmid construction to more precisely address this concern.

7) Minor comment: Figure 6 needs statistical tests.

-At the reviewers request we did perform ANOVA and Tukey-Kramer's HSD post-hoc test on this data. We have reported these results in the revised figure legend for figure 6.

Reviewer #2: 

1. Authors shouldn't refer to "dimerization null constructs", but rather "dimerization incompetent constructs" throughout the text.

-As suggested, we have made this change throughout the manuscript.

2. In Fig 1D, it would be good if authors could exactly specify the high/low affinity DNA sequences identified in a Supplementary Fig.

-As suggested, we have prepared a new more detailed figure S1 that shows the DNA sequences of all of the high/low affinity sites.

3. In the discussion, authors hypothesize that R1945 might be relevant in N4ID for dimerization. Can authors check this? This should be fairly easy to do using their reporter system, upon site-directed mutagenesis for that position.

-We agree with the reviewer that this is an experiment that needs to be done. At this time however, Mr. Crow is ready to leave the lab and the project has been passed onto a new student. This will be the subject of a future study.

4. Please correct grammar in the Discussion in the sentence "This indicated

that long-range interactions between NTC complexes are unlikely TO occur"

-done.

---

## [Decision Letter · Decision Letter 1]

16 Sep 2020

PONE-D-20-14835R1

Notch family members follow stringent requirements for intracellular domain dimerization at sequence-paired sites

PLOS ONE

Dear Dr. Albig,

Thank you for submitting your manuscript to PLOS ONE. After careful consideration, we feel that it has merit but does not fully meet PLOS ONE’s publication criteria as it currently stands. Therefore, we invite you to submit a revised version of the manuscript that addresses the points raised during the review process. Please pay particular attention to the comment regarding the evidence for N1ICD and N4ICD heterodimerization.

We look forward to receiving your revised manuscript.

Kind regards,

George Mosialos

Academic Editor

PLOS ONE

Reviewers' comments:

Reviewer's Responses to Questions

**Comments to the Author**

1. If the authors have adequately addressed your comments raised in a previous round of review and you feel that this manuscript is now acceptable for publication, you may indicate that here to bypass the “Comments to the Author” section, enter your conflict of interest statement in the “Confidential to Editor” section, and submit your "Accept" recommendation.

Reviewer #1: (No Response)

2. Is the manuscript technically sound, and do the data support the conclusions?

Reviewer #1: Partly

3. Has the statistical analysis been performed appropriately and rigorously? 

Reviewer #1: Yes

4. Have the authors made all data underlying the findings in their manuscript fully available?

Reviewer #1: Yes

5. Is the manuscript presented in an intelligible fashion and written in standard English?

Reviewer #1: Yes

6. Review Comments to the Author

Reviewer #1: Review of: “Notch family members follow stringent requirements for intracellular domain dimerization at sequence paired sites.”

Summary: In the revised version of this paper, the authors addressed several of my comments – in particular the fact that the reporter contained a minimal promoter (not promoterless), some additional data showing that the NICD-RA mutants could still activate gene expression, and re-wording of the abstract to not state that N4ICD forms dimers on SPS sites. However, one point was not addressed in a very convincing way and I also raise a few other minor issues the authors should consider.

1) I still do not believe the ChIP study demonstrates that N1ICD and N4ICD form heterodimers. The luciferase 2xTP1 DNA used in this assay has two Rbpj binding sites. Thus, when the ChiP-reChiP assay is done and the authors find that both N1ICD and N4ICD are bound to the same DNA – how do they know it forms a heterodimer (which is what they claim) versus simply one site bound by N4 and the other site independently bound by N1? That is why I recommended doing additional control experiments with Rbpj binding sites separated by 21 nts – which are non- cooperative sites. In fact, the authors themselves stated in the rebuttal to my comments the following: “since the 21bp promoter can still theoretically bind to two different tagged (but undimerized) NICD molecules, we would probably still detect ChIP on the 21bp promoter.” But that argument is also true of a 16bp spacer as well. I also recommended testing dimer-deficient NICD molecules in this assay – to which they stated in their rebuttal: “on the surface, this sounds like an excellent control for our experiment, however, it is necessary to remember that any two NICD molecules might be able to bind to the DNA regardless if they are dimer capable or incapable.” And yet, in the manuscript, the authors state that their data supports the conclusion that N1 and N4 form heterodimers on SPS sites. In my opinion, this data shows that N4 and N1 can both bind to the same DNA that has two Rbpj binding sites at the same time. It does not show that these molecules form heterodimers.

2) Figure 1B needs significance tests between the WT and RA mutants for N2ICD, N3ICD, and N4ICD.

3) A point of clarity – In Fig 4A and 4B the sequence is provided for the SPS site tested and it is labeled TP1 (complete) and TP1 (core). However in all the luciferase data it is called 2xTP1(SPS)-complete or 2xTP1(SPS)-core. Does that mean it contains two copies of the core and complete sequence? Or should the above sequence be re-labeled as 2xTP1?

4) A minor comment for the authors to consider – but it is their choice to leave it or change it. I still personally find the small differences in luciferase activity between the WT and RA mutant NICD molecules to be overstated. Hence, I don't really understand why they make such a big deal about these small differences (luciferase assays are really sensitive and it is unclear what a less than a 2-fold difference means). Hence, I find their argument in the section on “Non-optimal SPS sites select against transcriptional activation by NICD dimers” to be less than compelling. But again, I am a firm supporter of the authors telling the story how they want to – I would just state that as a person that has studied transcription for over 25 years and performed luciferase assays throughout that time – it is really hard to determine the importance of such small changes, even when statistically different.

7. PLOS authors have the option to publish the peer review history of their article (what does this mean?). If published, this will include your full peer review and any attached files.

Reviewer #1: No

---

## [Author Response · Author response to Decision Letter 1]

27 Oct 2020

10-26-2020

Dear Dr. Mosialos,

Thank you for continuing serving as editor for our submission to PLOS One. We again thank reviewer #1 for his/her thoughtful review of our manuscript. I must say that I admire reviewer #1 for the dedication shown in reviewing our manuscript. In the true sense of the peer review process, our manuscript has been greatly improved due to the reviewer's diligence. I hope our new revisions will be satisfactory. We have addressed the concerns below.

1) I still do not believe the ChIP study demonstrates that N1ICD and N4ICD form heterodimers. The luciferase 2xTP1 DNA used in this assay has two Rbpj binding sites. Thus, when the ChiP-reChiP assay is done and the authors find that both N1ICD and N4ICD are bound to the same DNA - how do they know it forms a heterodimer (which is what they claim) versus simply one site bound by N4 and the other site independently bound by N1? That is why I recommended doing additional control experiments with Rbpj binding sites separated by 21 nts - which are non- cooperative sites. In fact, the authors themselves stated in the rebuttal to my comments the following: "since the 21bp promoter can still theoretically bind to two different tagged (but undimerized) NICD molecules, we would probably still detect ChIP on the 21bp promoter." But that argument is also true of a 16bp spacer as well. I also recommended testing dimer-deficient NICD molecules in this assay - to which they stated in their rebuttal: "on the surface, this sounds like an excellent control for our experiment, however, it is necessary to remember that any two NICD molecules might be able to bind to the DNA regardless if they are dimer capable or incapable." And yet, in the manuscript, the authors state that their data supports the conclusion that N1 and N4 form heterodimers on SPS sites. In my opinion, this data shows that N4 and N1 can both bind to the same DNA that has two Rbpj binding sites at the same time. It does not show that these molecules form heterodimers.

Originally, this figure was only intended to show that the 2xTP1(core) construct was in fact able to bind NICD molecules, but it has (for the better) taken on a whole new "life of its own". The reviewer is 100% correct about the concerns with the experiment and we have performed the suggested experiments. I have included the new data in this letter. The result is very interesting and raises new questions about how Notch functions. However, as is so often the case, this new data raises more questions than it answers. And so, given that the manuscript does not hinge on this data, I have opted to remove panel 4C from the final manuscript. That said, I plan to pursue this ChiP approach to see what it can teach us (if anything) about how Notch functions. This data can be found in the submitted response to reviewers letter.

The new data shows (as the reviewer correctly guessed) that NICD molecules can associate with the DNA even if the gap between RBPj binding sites is non-optimal (21 bp) or the NICD molecules are dimer-null (Ank Mut). Keep in mind however that the N4ICD ankyrin mutation used here is the same mutation we used in the manuscript that was shown to NOT break N4ICD dimerization activity (I have a student currently trying to solve that little N4ICD mystery). However, the N1ICD ankyrin mutation does certainly suppress transcription from SPS sites with 16bp gaps as shown in our manuscript. Regardless, we are forced to agree with the reviewer that this data cannot distinguish between 1.) NICDs engaging in dimerization on the DNA, or 2.) two NICD molecules binding to DNA independently of dimerization.

That said, the new ChiP data does raise other questions and I would welcome the reviewers input about what this data might mean. If the reviewer feels comfortable, I would welcome a discussion in any format he/she feels appropriate with. Even as an anonymous note through the official reply to this resubmission would be welcome. In particular, the new data is particularly interesting since we consistently recover the most DNA in the 4/4 sample ChiP sample, followed by the 1/4 sample, followed by 1/1 sample. Almost always in that order. And, it is not just the transfected target DNA since ChiP analysis of the chromosomal Hes1 and Hes4 promoters returned the same pattern of DNA recovery (not shown). 

If this data does not illustrate dimerization, what does it mean? Does this data suggest that N4ICD simply binds to RBPj and DNA better than N1ICD? Does it suggest an activity of NICD binding to DNA that we are not aware of? Or, maybe the data is screaming the obvious at us … maybe NICDs bind just fine to 21 bp gapped SPS elements and they don't need ankyrin domains to bind (this is what the data says). Maybe the only way to get the synergistic amplification of transcription that comes with NICD dimerization is to have dimerization on the 16bp gap so that MAML and p300 can bind to the a NICD complex with the correct dimensions? My student's and I are currently scratching our heads over this one. In the end, since the data is not critical for the overall paper, I have decided to simply remove the data and save it for another time when we can more robustly analyze the experiment, perform follow up experiments, and give these questions the full attention they deserve.

2) Figure 1B needs significance tests between the WT and RA mutants for N2ICD, N3ICD, and N4ICD.

This has been updated as requested.

3) A point of clarity - In Fig 4A and 4B the sequence is provided for the SPS site tested and it is labeled TP1 (complete) and TP1 (core). However in all the luciferase data it is called 2xTP1(SPS)-complete or 2xTP1(SPS)-core. Does that mean it contains two copies of the core and complete sequence? Or should the above sequence be re-labeled as 2xTP1?

Thank you for pointing this out. These are both 2xTP1 elements and we have modified the description of each DNA sequence appropriately.

4) A minor comment for the authors to consider - but it is their choice to leave it or change it. I still personally find the small differences in luciferase activity between the WT and RA mutant NICD molecules to be overstated. Hence, I don't really understand why they make such a big deal about these small differences (luciferase assays are really sensitive and it is unclear what a less than a 2-fold difference means). Hence, I find their argument in the section on "Non-optimal SPS sites select against transcriptional activation by NICD dimers" to be less than compelling. But again, I am a firm supporter of the authors telling the story how they want to - I would just state that as a person that has studied transcription for over 25 years and performed luciferase assays throughout that time - it is really hard to determine the importance of such small changes, even when statistically different.

I completely understand your point here and I hesitate to fly in the face of experience. It is not a major point in our paper, however I think that I would like to keep this data in the paper. My reason for doing this is that there is not much known about how the NICD complexes actually bind DNA and initiate transcription. My gut instinct is telling me that this data might be able to teach us something about Notch. Hopefully, it will be useful for others.

---

## [Decision Letter · Decision Letter 2]

10 Nov 2020

Notch family members follow stringent requirements for intracellular domain dimerization at sequence-paired sites

PONE-D-20-14835R2

Dear Dr. Albig,

We’re pleased to inform you that your manuscript has been judged scientifically suitable for publication and will be formally accepted for publication once it meets all outstanding technical requirements.

Kind regards,

George Mosialos

Academic Editor

PLOS ONE

Additional Editor Comments (optional):

Reviewers' comments:

Reviewer's Responses to Questions

**Comments to the Author**

1. If the authors have adequately addressed your comments raised in a previous round of review and you feel that this manuscript is now acceptable for publication, you may indicate that here to bypass the “Comments to the Author” section, enter your conflict of interest statement in the “Confidential to Editor” section, and submit your "Accept" recommendation.

Reviewer #1: All comments have been addressed

2. Is the manuscript technically sound, and do the data support the conclusions?

Reviewer #1: Yes

3. Has the statistical analysis been performed appropriately and rigorously? 

Reviewer #1: Yes

4. Have the authors made all data underlying the findings in their manuscript fully available?

Reviewer #1: Yes

5. Is the manuscript presented in an intelligible fashion and written in standard English?

Reviewer #1: Yes

6. Review Comments to the Author

Reviewer #1: The authors have done a great job addressing the remaining concerns and I strongly support publication.

Regarding the request for my thoughts on the new unpublished ChIP-reChIP data - I don't have anything profound to say! One thing that I also struggle with is the association of amount of DNA binding with the amount of transcriptional activation. In some cases, there has been a clear link between transcription factor turnover and the amount of transcription activation with higher rates of protein turnover linked with increased transcription. So, one possibility is that NICD1 may be more rapidly phosphorylated and ubiquitylated when bound to SPS sites and thus turnover quickly whereas NICD4 may have a slow turnover kinetics but not activate transcription as well...... Again, totally my guess but there has been a recent paper in eLife with some evidence that NICD binding in flies leads to faster turnover when bound to SPS sites than CSL sites - however, this phenomenon was not linked directly to transcriptional activation.

Best of luck on your future work!

7. PLOS authors have the option to publish the peer review history of their article (what does this mean?). If published, this will include your full peer review and any attached files.

Reviewer #1: **Yes: **Brian Gebelein

---

## [Editor Report · Acceptance letter]

13 Nov 2020

PONE-D-20-14835R2 

Notch family members follow stringent requirements for intracellular domain dimerization at sequence-paired sites 

Dear Dr. Albig:

I'm pleased to inform you that your manuscript has been deemed suitable for publication in PLOS ONE. Congratulations! Your manuscript is now with our production department. 

Kind regards, 

on behalf of

Dr. George Mosialos 

Academic Editor

PLOS ONE